# Nutrient Solution Electrical Conductivity Affects Yield and Growth of Sub-Irrigated Tomatoes



**Ariel Méndez-Cifuentes** [1] , **Luis Alonso Valdez-Aguilar** [2,*], **Martín Cadena-Zapata** [1],
**Daniela Alvarado-Camarillo** [3] **and José Antonio González-Fuentes** [2]

1   Departamento de Maquinaria Agrícola, Universidad Autónoma Agraria Antonio Narro,
    Calzada Antonio Narro 1923, Buenavista, Saltillo 25315, Mexico; mendezc.ariel@gmail.com (A.M.-C.);
    martin.cadena@uaaan.edu.mx (M.C.-Z.)
2   Departamento de Horticultura, Universidad Autónoma Agraria Antonio Narro, Calzada Antonio Narro 1923,
    Buenavista, Saltillo 25315, Mexico; jagf252001@gmail.com
3   Departamento de Ciencias del Suelo, Universidad Autónoma Agraria Antonio Narro,
    Calzada Antonio Narro 1923, Buenavista, Saltillo 25315, Mexico; daniela.alvaradoc@uaaan.edu.mx
*   Correspondence: luisalonso.valdez@uaaan.edu.mx

**Abstract:** Sub-irrigation of greenhouse crops has the potential to increase water and nutrient use efficiency; however, fertilizer salts that are not absorbed by the plants tend to accumulate in the substrate and eventually raise the substrate's electrical conductivity (EC). The objective of this study was to determine the optimum EC of the nutrient solution in sub-irrigated tomatoes to allow maximum yield. Total fruit yield was higher in sub-irrigated plants with solutions at 2.0 dS m$^{-1}$ (5105 g per plant), and it was comparable to that obtained for drip-irrigated plants (4903 g per plant); however, the yield of fruits from the second truss was 37% higher in sub-irrigated than in drip-irrigated plants when the EC was 2.0 dS m$^{-1}$. In contrast, at the end of the growing season, the yield of plants sub-irrigated with nutrient solutions of 2.0 dS m$^{-1}$ was the lowest, being surpassed by 37% by that of plants treated with 1.4 dS m$^{-1}$. The dry weight of vegetative plant parts was reduced in sub-irrigated plants, suggesting a shift in dry mass partitioning. Our results show that with sub-irrigation, the growing season should be started using nutrient solutions with higher EC, but eventually, this EC should be decreased to maintain proper substrate EC and high yield.

**Keywords:** closed-loop irrigation; ebb and flow; containerized crops; fertilizer and water use efficiency

## 1. Introduction

Increasing shortages of good quality water for irrigation purposes is challenging the long-term sustainability and resilience of agricultural production [1]. In addition, ongoing disruptions in fertilizer production and subsequent supply chains issues driven by political issues have resulted in a marked increase in input costs, which continue to challenge food production [2]. Soilless production practices have the potential to reduce fertilizer inputs and water loss, thereby offering the means for effective and environmentally friendly production of food [3–5]. However, there is concern as to the sustainability of soilless/hydroponic production systems, as they depend on excessive use of synthetic fertilizers to prepare nutrient solutions [6], which may cause harm to the environment if not correctly disposed of or treated [7].

Greenhouses are increasingly designed with irrigation systems that are capable of delivering nutrient solutions to plants to maximize growth and yield, and capture and recycle the nutrient solution that is not retained by the substrate for reuse in subsequent irrigation events [8]. Ebb and flow sub-irrigation systems have been widely used for growing potted ornamental plants, and there is increasing interest in taking advantage of these closed-loop designs to reduce nutrient and water loss to the environment [9] during the production of fruit and vegetables.

According to Ferrarezi et al. [4], sub-irrigation is a technique that provides fertilizer solution to the bottom of containers, and through capillary action, water and nutrients are provided to the roots through holes located in the container. In this system, container-grown plants are periodically flooded with a nutrient solution using a closed-loop system, which collects and reuses any excess nutrient solution for subsequent irrigation events.

In open irrigation systems, fertilizer salts not absorbed by the plants do not accumulate in the substrate due to the high lixiviation rate; however, this system has a very high environmental and economic cost, as the nutrients and water are frequently allowed to drain and accumulate in the surrounding environment [10,11]. In contrast, in closed-loop sub-irrigation systems, non-absorbed fertilizer salts accumulate in the upper portion of the root ball, where fewer roots are located [12]; the reduced lixiviation in this system results in accumulation of the excess of fertilizers and non-absorbed salts in the upper portion of the substrate which eventually may raise the electrical conductivity (EC) of the substrate [4,13–15] and negatively affect plant growth and yield. For example, in sub-irrigated tomatoes, the EC in the top layer of the substrate has been reported to be as high as 15.5 dS m$^{-1}$ at the end of the growing season [16].

It has been reported that to reduce the deleterious impact of increasing substrate EC associated with sub-irrigation, nutrient solution should have a lower concentration, and thus a lower EC, than that used in conventional drip-irrigation techniques [17]. Giuffrida and Leonardi [18] reported that using reduced nutrient concentrations in a closed-loop soilless system had no significant effect on fruit yield but improved the use of water and minerals in pepper (*Capsicum annuum* L.). However, El Youssfi et al. [19] reported that half-strength nutrient solution concentration in beans (*Phaseolus vulgaris* L.) may still be excessive, as it resulted in a 15% reduction in yield.

Sub-irrigation in soilless containerized crop production has many advantages, including a higher water use efficiency; for example, it has been demonstrated that 1 kg of fresh tomatoes can be produced with 22 L of water in a sub-irrigation system, compared to the 41 L required in a drip-irrigated system [20]. It has also been reported that 87% and 219% less water are required for the production of citrus liners [21] and forestry trees nursery [22], respectively, when plants were sub-irrigated, compared to overhead hand-watered plants. The enhanced water use efficiency via sub-irrigation has also been associated with higher nutrient use, as demonstrated in the early generative phase of chrysanthemum (*Chrysanthemum morifolium* Ramat) since plants were able to increase the uptake of N and K at a higher rate compared to the concentration of these elements in the nutrient solution [23]; in tomatoes, sub-irrigation resulted in a significantly lower nutrient supply being enough to meet plant demands with no negative impacts on fruit yield [20]. This substantial improvement in the use of nutrients and water in sub-irrigation systems with nutrient solution recirculation makes it necessary to evaluate different ECs, calculated on a nutrient concentration basis, to minimize the impact on the substrate's EC due to the salts accumulated in the substrate without provoking nutrient deficiencies. Therefore, this study was conducted to determine the effect a nutrient solution EC on tomato fruit production and plant growth in a soilless containerized sub-irrigation system when compared to conventional drip-irrigation.

## 2. Materials and Methods

### 2.1. Growing Conditions

The study was conducted in a greenhouse located at the Universidad Autónoma Agraria Antonio Narro, Saltillo, Coahuila, México (25°21′24″ N, long. 101°02′05″ W). For temperature control, the greenhouse had a fan and pad cooling system with a temperature set point of 25.0 °C and a heater unit set at 10 °C. The mean temperature during the experiment was 20.1 °C (average minimum 14.4 °C and maximum 25.7 °C), and the average relative humidity was 68% (average minimum 43% and average maximum 92%). The average photosynthetic photon flux density was 389 μmol m$^{-2}$ s$^{-1}$.

## 2.2. Plant Material

Seedlings of hybrid tomato (*Solanum lycopersicum* L.) cv Climstar were transplanted on 7 August 2018 and the experiment was concluded on 11 January 2019. At transplant, the seedlings were 20 cm in height, had two fully expanded leaves and were planted in 10 L black-color plastic pots (one plant per pot) filled with a mixture of sphagnum moss (40% *v/v*) (PREMIER, Premier Tech, Toronto, Canada), coir (40% *v/v*) (Germinaza, Colima, México), and perlite (20% *v/v*) (HORTIPERL, Termolita, Monterrey, México). The initial pH and EC of the substrate were adjusted to 5.8 and 0.2 dS m$^{-1}$, respectively. Plants were trellised to one stem by removing the side shoots, while the leaves were pruned periodically throughout the study period to maintain 11 to 13 mature leaves; eight trusses were allowed to develop and they were pruned to maintain five flowers each.

## 2.3. Sub-Irrigation and Drip-Irrigation Set Up

The sub-irrigation consisted of rigid plastic trays of 69 cm length, 39 cm width, and 16 cm height. In each tray, two 1-plant pots were placed 30 cm apart within rows and the rows were 120 cm apart. Each tray had a net of polyvinyl chloride pipes and valves for sub-irrigation with the nutrient solution with the corresponding EC; each nutrient solution was pumped with a $\frac{1}{4}$ HP pump. Tomato plants were sub-irrigated when the substrate had a moisture tension of 10 KPa (Irrometer Model MLT, Riverside, CA, USA). Irrigation halted when the nutrient solution reached a depth of 8 cm of the tray and the containers remained stranded on the nutrient solution for 30 min to allow capillary rise through the substrate. The solution that was not absorbed by the substrate was returned into a 200 L cistern for reuse in the following irrigation. The nutrient solutions were replenished to compensate evapotranspiration of water and renovated every 15 days. Prior to irrigation, the nutrient solutions were adjusted for pH to 6.0 ± 0.1 with 0.1 N H$_2$SO$_4$. For comparison purposes, a set of plants were watered with a drip-irrigation system; on each pot, two 2 L h$^{-1}$ emitters were placed; drip irrigation was also conducted at 10 KPa and halted when a 30% leaching fraction was accomplished.

## 2.4. Electrical Conductivity Treatments

Three ECs of the nutrient solution were assessed in this experiment. The nutrient solution at 2.0 dS m$^{-1}$ had the following composition: 14 meq L$^{-1}$ NO$_3$$^-$, 2 meq L$^{-1}$ H$_2$PO$_4$$^-$, 8 meq L$^{-1}$ SO$_4$$^{2-}$, 11 meq L$^{-1}$ Ca$^{2+}$, 9 meq L$^{-1}$ K$^+$ and 4 meq L$^{-1}$ Mg$^{2+}$; the other ECs, 1.4 and 2.4 dS m$^{-1}$, were calculated by reducing or augmenting the concentrations of the ions by 30% and 20%, respectively. The pH of the nutrient solution ranged between 5.8 and 5.9. The EC of the nutrient solution for drip-irrigated plants was 2.0 dS m$^{-1}$. Micronutrients were provided at 5 mg L$^{-1}$ Fe (Fe-EDTA), 2.5 mg L$^{-1}$ Mn (Mn-EDTA), 0.25 mg L$^{-1}$ B, 0.4 mg L$^{-1}$ Zn (Zn-EDTA) and 0.2 mg L$^{-1}$ Cu (Cu-EDTA).

## 2.5. Yield and Growth Parameters

Fruit harvest started 10 weeks after transplanting. Fruits were considered ripe when 90% of the characteristic color of the cultivar was displayed. The yield of fruits for each of eight trusses (inflorescences) harvested was recorded and at the study's termination, the total yield was calculated. At the experiment's termination, the stem, leaves, and fruits were collected and dried at 70 °C; after 72 h, the plant parts were weighed in an analytical balance (VELAB VE-1000, Ciudad de México, México). All the harvested fruits from each of the four treatments were classified according to the US Consumer Standards for Fresh Tomatoes [24] into small (<85.0 g), medium (85.0 to 170.1 g), large (170.1 to 283.5 g), and very large (>283.5 g), and the percent of each category was calculated per truss and on a total basis.

## 2.6. Substrate pH and EC

At the study's termination, EC and pH were measured in substrate samples drawn from the middle layer of the root ball using the 1:2 dilution method; one part of the

substrate sample was mixed with two parts of water and allowed to sit for 60 min, then the pH and EC were measured with a portable pH meter and conductivity meter (Horiba LAQUA Twin, Kyoto, Japan). However, for sub-irrigated plants, the EC of the substrate was recorded daily with sensors placed in the middle section of the container at a 7.5 cm depth; the sensors were connected to a data logger Em50 series as part of the system $ECH_2O$ (Decagon Devices, Pullman, WA, USA).

### 2.7. Statistical Analysis

The experimental unit for the sub-irrigated and the drip-irrigated treatments was two plants. A completely randomized blocks design was used to establish the experiment. The study consisted of three EC treatments in sub-irrigation (1.4, 2.0, and 2.4 dS m$^{-1}$) and a set of drip-irrigated plants as a control whose nutrient solution was of 2.0 dS m$^{-1}$; each treatment had six 2-plant replications. The data obtained were analyzed with an analysis of variance with R Studio version 3.4.2 [25] and the means were compared through Tukey's procedure ($p \leq 0.05$) when significance was detected.

## 3. Results and Discussion

### 3.1. Plant Growth

Plants sub-irrigated with solutions at 2.4 dS m$^{-1}$ showed a significant biomass reduction as the stem and leaf dry weight were lower than those of drip-irrigated plants, while the stem or leaves biomass decreased when plants were sub-irrigated with solutions of 1.4 and 2.0 dS m$^{-1}$, respectively (Figure 1); nonetheless, in sub-irrigated plants, the stem, leaves, and total dry weights were similar for all the ECs assessed. These results are in contrast with reports showing that increasing the EC of the nutrient solution results in a lineal raising of the aerial and root biomass in sub-irrigated tomato [16]. We suggest that these differences could be because, in the present study, the ECs evaluated were higher than those used in previous studies. The decreased biomass accumulation in the plant parts of tomato sub-irrigated with nutrient solutions of the lowest EC may be ascribed the reduction in nutrient concentration, suggesting that the nutrient demands by tomato plants were not met; this is in agreement with Heinen et al. [26] who reported that at low EC there are not enough nutrients available, resulting in a reduction of nutrient uptake and crop growth. Our results concur with reports by Rouphael and Colla [27] for sub-irrigated pumpkin (*Cucurbita pepo* L.) as the dry weight declined when the EC (or concentration) of the nutrient solution was reduced to 50%, leading to insufficient nutrient supply. In rockwool cultures, tomato plants also exhibited higher fresh weight of plant parts at 2.0 dS m$^{-1}$, but higher or lower ECs resulted detrimental for biomass production [26].

### 3.2. Fruit Yield

The total yield of fruits was highest when plants were sub-irrigated with solutions of 2.0 dS m$^{-1}$; this yield was comparable to that obtained via drip irrigation and to that of plants sub-irrigated with solutions of 1.4 dS m$^{-1}$ (Figure 2). However, increasing EC to 2.4 dS m$^{-1}$ caused a significant 18% decrease in the yield compared to that obtained when it was 2.0 dS m$^{-1}$ (Figure 2). Our results concur with previous reports indicating that yield in tomato was the highest when the nutrient solution had an EC of 2.0 dS m$^{-1}$ [16, 28], and similar to other reports indicating significant decreases in yield and water use efficiency when applying 1.5× and 2× the strength of the nutrient concentration [13]. Nonetheless, our results contrast with reports indicating that, compared to drip-irrigated plants, the tomato exhibit yields reductions from 13.6% to 19.7%, due to the effect of sub-irrigation [15]; other studies have also demonstrated negative effects on yield associated with sub-irrigation in vegetable species, for example, in zucchini, there was a 58% [27] or 18% [29] reduction in the yield when the EC of the nutrient solution was decreased by 50%, whereas in sub-irrigated green beans there was a 33% decrease in fruit production [30]. The contrasting responses to the sub-irrigation could be due to each plant species' tolerance to high EC and the specific demand for nutrients [4,31]. In our study, the decrease in

fruit production with solutions of 2.4 dS m$^{-1}$ may be associated with the accumulation of fertilizer salts in the substrate, causing a marked increase in the EC and therefore affecting plants by increasing the osmotic stress, the damage on membrane integrity, and the impaired water uptake and nutrient imbalance [32,33]. Reports indicate that in drip-irrigated tomato, a threshold EC for the nutrient solution has been established at 3.2 dS m$^{-1}$, after which tomato exhibits significant yield reduction [34]. Fayezizadeh et al. [35] and Heinen et al. [26] indicated that if the concentration of nutrients or the EC in the nutrient solution exceeds the optimal range, it reduces the growth and yield of plants in closed hydroponic systems as they are under more severe osmotic conditions.

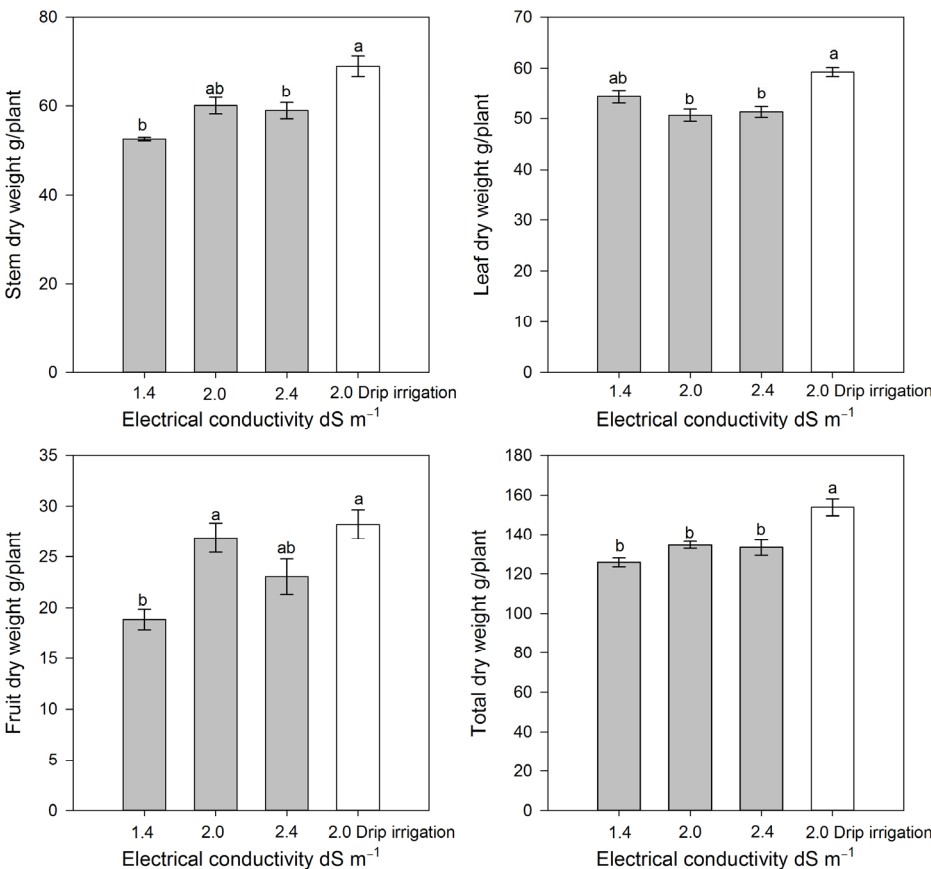

**Figure 1.** Biomass of tomato plants sub-irrigated (gray columns) or drip-irrigated (white column) with nutrient solutions of different electrical conductivities. Bars are the standard error of the mean. Different letters indicate significant differences according to Tukey´s multiple mean comparison test (*p* < 0.05).

According to Wang et al. [36] the ebb and flow sub-irrigation technique is a promising strategy to promote the growth of vegetable crops. Our results concur with that statement, as they showed that tomatoes may be grown under sub-irrigation with no detrimental effects on fruit yield as long as the EC of the nutrient solution does not exceed 2.0 dS m$^{-1}$. When the EC of the nutrient solution was 2.0 dS m$^{-1}$, the yield of sub-irrigated plants was comparable to that of drip-irrigated plants. This is in line with reports indicating that, if the increase in nutrient solution concentration is within the adequate range for tomato, the fruit yield is not significantly affected, but if the concentration is increased to very high proportions it will reduce crop yield [35]. It has been suggested that some sub-irrigation systems are not recommended for 'slow process' plants or plants that require more water [10]. In greenhouse tomato production, the growing season is frequently extended for a long period of time (up to 8–10 months), which increases the exposure of the crop to the deleterious effects of increased substrate EC due to the upward migration of salts in the substrate overtime. In contrast, ornamental potted plants, such as geranium

(*Pelargonium* × *hortorum* Bailey) [37–39] and chrysanthemum [39], that are usually grown in the ebb and flow sub-irrigation systems, have a shorter growing season, 60 to 90 days after transplant, so that exposure to the potential deleterious effect of the high substrate EC is reduced.

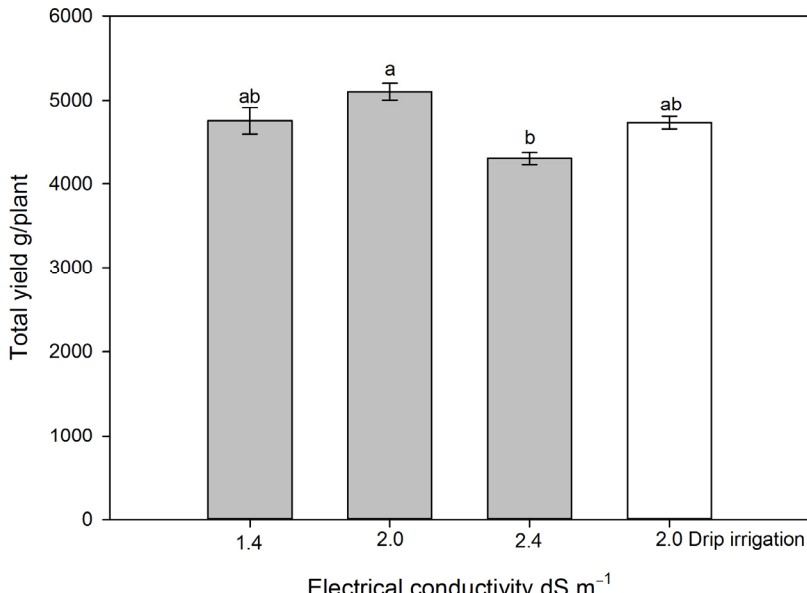

**Figure 2.** Total fruit yield in sub-irrigated (gray columns) and drip-irrigated (white column) tomato plants as affected by the electrical conductivities of the nutrient solution. Bars are the standard error of the mean. Different letters indicate significant differences according to Tukey´s multiple mean comparison test ($p < 0.05$).

Our results indicate that unaffected fruit production in sub-irrigated plants was associated with a decrease in biomass of the vegetative plant parts (Figure 1), suggesting that under sub-irrigation there is a tendency to divert the biomass towards fruit formation, as yield was not affected. The fact that plants sub-irrigated with nutrient solutions of 1.4 and 2.0 dS m$^{-1}$ exhibited a yield comparable to that of the drip-irrigated plants despite their lower biomass accumulation suggests that tomato partitioning of reserves were shifted towards fruit production; similar trends have been reported by Méndez-Cifuentes et al. [20]. According to Ji et al. [40] dry mass partitioning to tomato fruits may be associated with an increased 'fruit-sink' strength due to enhanced transport and metabolism of sugars, which in turn may be related to an improved nutrient balance.

*3.3. Fruit Yield Evolution*

The effect of the EC of the nutrient solution on yield was not consistent throughout the growing season, as it varied according to the truss from which the tomatoes were harvested (Figure 3). The yield of tomatoes harvested from the second and fourth trusses was higher when plants were sub-irrigated with a nutrient solution of EC at 2.0 dS m$^{-1}$ compared to that of plants sub-irrigated with solutions at 1.4 and 2.4 dS m$^{-1}$ or drip-irrigated plants (Figure 3). Nonetheless, at the study's termination, tomatoes from the eighth truss of plants irrigated with the nutrient solution of the lowest EC had the highest yield (+37% higher yield than that of plants with solution at 2.0 dS m$^{-1}$). These results indicate that in order to avoid yield reduction over time in sub-irrigated tomato, the concentration of the nutrient solution must be adjusted as the growing season progresses. García-Santiago et al. [16] reported similar tendencies as during the first month of harvest, sub-irrigated tomatoes rendered the highest yields when the nutrient solution had an EC of 1.6 and 2.0 dS m$^{-1}$, while from the fourth to the sixth month, the higher yields were from plants treated with nutrient solution at 1.2 dS m$^{-1}$. On the first, third, fifth, sixth, and seventh truss there were

no effects on yield among ECs (Figure 3). Fayezizadeh et al. [35] reported similar results as fruits from the first to the third cluster in tomato cv. V4-22 produced yields from 548 to 634 g in a closed-loop irrigation system; however, the yield decreased gradually up to 394 g by the seventh cluster.

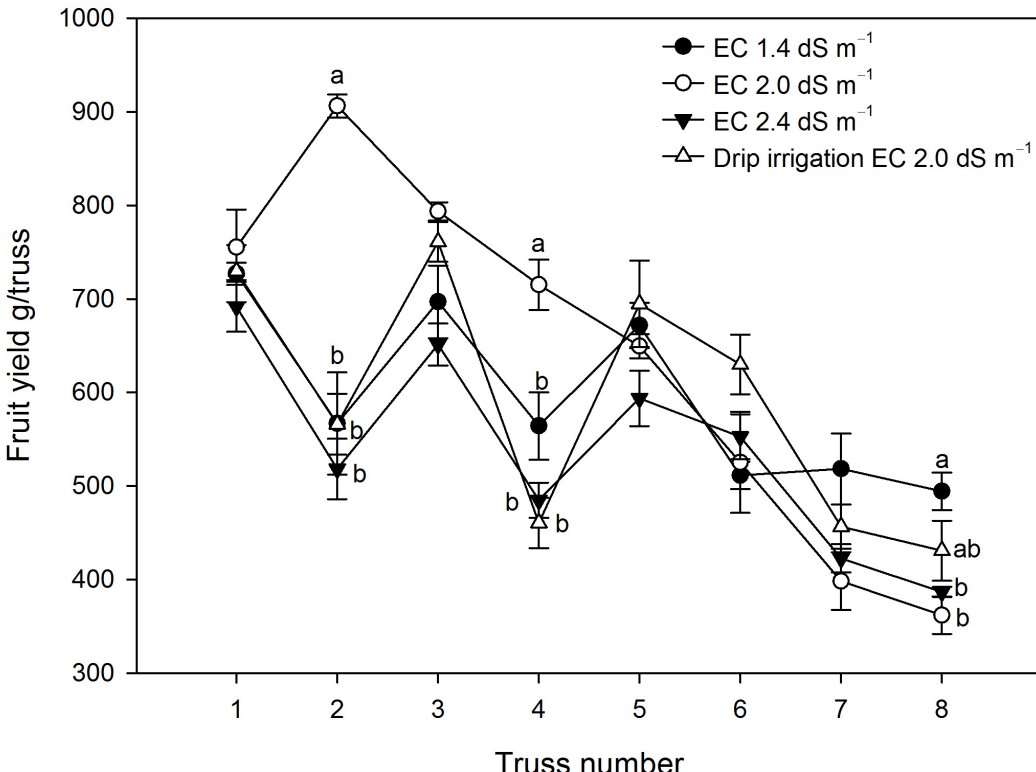

**Figure 3.** Fruit yield from eight tomato trusses as affected by the electrical conductivity (EC) of the nutrient solution in sub-irrigated or drip-irrigated plants. Bars are the standard error of the mean. Different letters indicate significant differences according to Tukey´s multiple mean comparison test ($p < 0.05$).

*3.4. Fruit Grades*

Considering all the fruits harvested throughout the growing season, drip-irrigated plants produced 48.5% and 31.2% of the fruits of medium and large size, respectively (Figure 4), which was comparable to that obtained by plants sub-irrigated with solutions of 2.0 dS m$^{-1}$ (51.8% and 31.3%, respectively). The percentage of small fruits was similar in plants from both treatments (~13.6%) (Figure 4). Sub-irrigating the plants with solutions of higher EC markedly increased the production of small fruits, at the expense of the percentage of large ones, whereas a lower EC was associated with higher production of medium-sized fruits (Figure 4). Wortman [6] reported similar tendencies, as the marketable yield of cherry tomatoes was 42% higher when the plants were cultivated in an ebb and flow sub-irrigation system with recirculation of the nutrient solution when the EC was up to 2.2 dS m$^{-1}$, compared to that of plants irrigated with a nutrient solution with EC of up to 1.0 dS m$^{-1}$. Magán et al. [34] also reported similar results as increasing the EC of the nutrient solutions in drip-irrigated tomatoes caused an increase in the percentage of fruits of small size, whereas fruits were predominantly of larger sizes when EC was low.

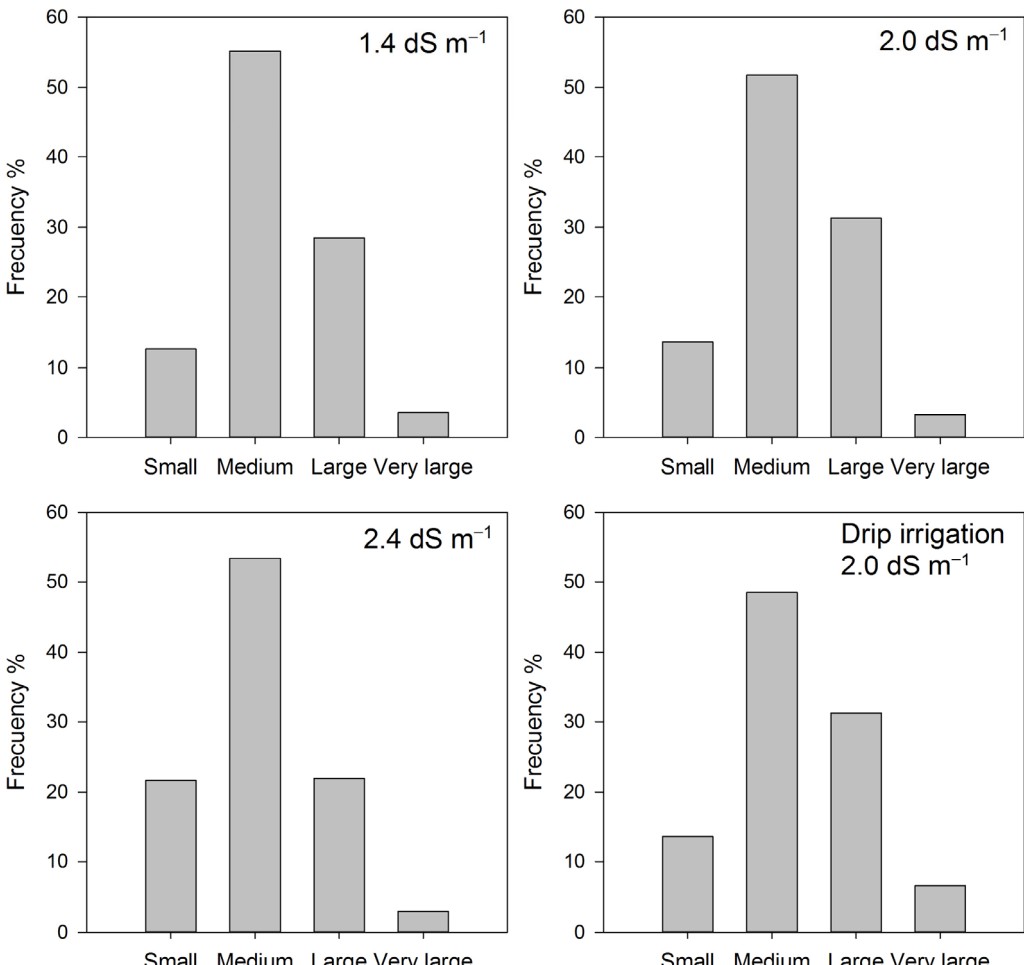

**Figure 4.** Frequency of tomato fruits graded according to the US Consumer Standards for Fresh Tomatoes [24] as affected by the electrical conductivity of the nutrient solution in sub-irrigated or drip-irrigated plants.

Nonetheless, as observed for fruit yield, fruit size varied according to the truss from which they were harvested. Fruits harvested from the first three trusses were predominantly medium to large in drip-irrigated plants, whereas those from the last two trusses were predominantly of medium size (Figure 5). In contrast, in plants sub-irrigated with nutrient solutions of 2.0 dS m$^{-1}$, the first six trusses produced predominantly fruits of medium and large size, so that the higher quality of fruits was maintained for a longer period of time; however, the last two trusses produced fruits predominantly of medium and small size due to a sharp increase in the latter as this went from 11.6% in the sixth truss to 46.8% in the eighth truss (Figure 5). In plants treated with a nutrient solution of 1.4 dS m$^{-1}$, fruits were also of medium and large size during the first six trusses, but at the end of the season, the production of large fruits declined while that of medium fruits rose (Figure 5). On nutrient solutions of higher EC (2.4 dS m$^{-1}$), most of the fruits were of medium size throughout the growing season, although during the last two trusses, the percentage of small fruit was similar or higher to that of medium size fruits (Figure 5).

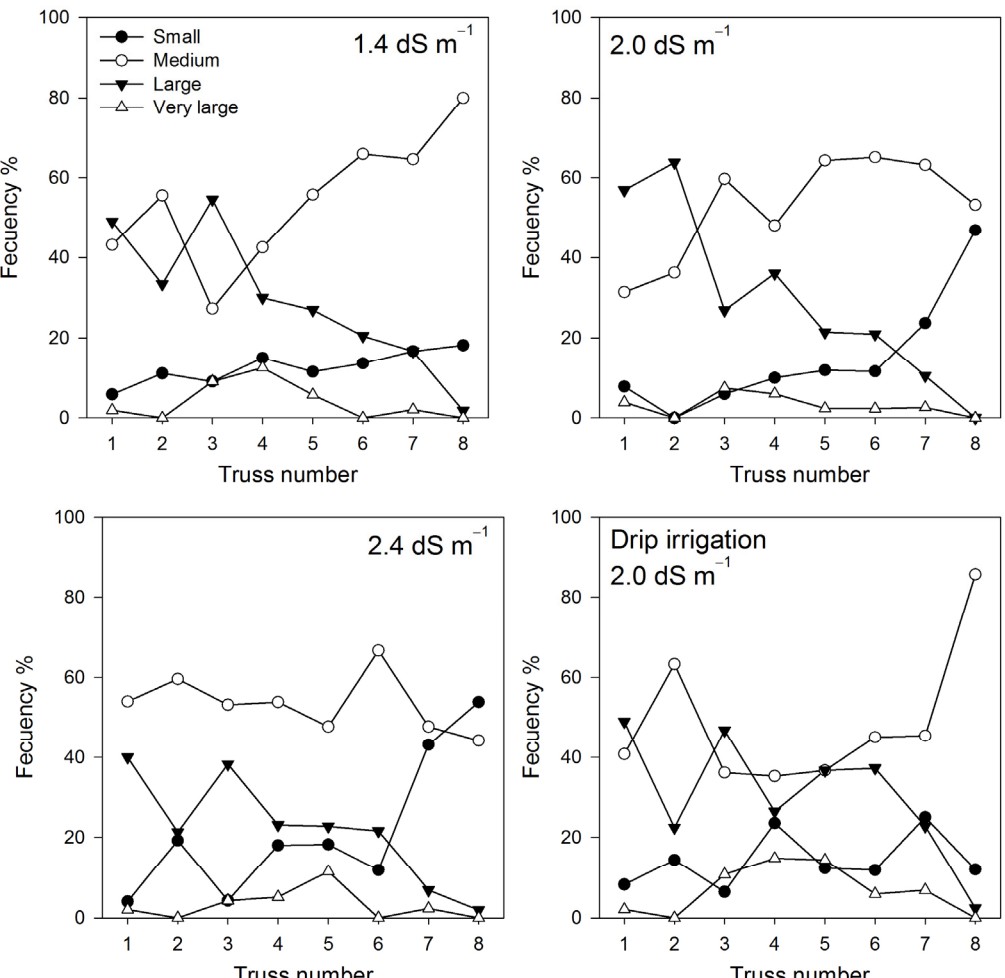

**Figure 5.** Frequency of tomato fruits graded according to US Consumer Standards for Fresh Tomatoes [24] from eight trusses as affected by the electrical conductivity of the nutrient solution in sub-irrigated or drip-irrigated plants.

In summary, our results indicate that to produce a fruit comparable in quality to that of drip-irrigated plants, tomato should be sub-irrigated with solutions of 2.0 dS $m^{-1}$, however, it is important to note that the last two trusses will produce a higher percentage of small fruits (Figure 5). A strategy to avoid the rise of small fruits at the end of the season may focus on reducing the EC of the nutrient solution further, to ~1.4 dS $m^{-1}$, as with this treatment, there was no increase in production of small fruits.

### 3.5. Substrate Electrical Conductivity and pH

At the study's termination, the substrate EC increased by 31% and 60% when the plants were sub-irrigated with the nutrient solution at 2.0 and 2.4 dS $m^{-1}$, respectively, when compared to that at 1.4 dS $m^{-1}$; however, the substrate of drip-irrigated plants showed the lowest EC (Table 1). Similar trends were reported in *Rhododendron yakushimanum* by Matysiak and Bielenin [41] indicating that, irrespective of the concentration of the nutrient solution, the resulting EC was higher in containers fertigated by sub-irrigation than those by overhead irrigation. The increase in EC under sub-irrigation is attributed to the accumulation of salts due to the upward water movement carrying the dissolved fertilizers by mass flow [8]. The increases in substrate EC and the decrease in pH may explain the decrease in fruit yield shown in Figure 2. The yield reduction of fruits with nutrient solutions of high EC is reported to be caused by the increase in substrate EC due to salt accumulation [42–44], which mainly occurs on the top layer of the substrate,

reaching up to 19.5 dS m$^{-1}$ when nutrient solutions of high EC are applied through sub-irrigation [16]; however, it has also been pointed out that the highest salinity at the substrate top layer may not affect plants' growth, since the roots develop predominantly from the middle to the bottom layers [15,45].

**Table 1.** Effect of the electrical conductivity (EC) in the nutrient solution on the substrate pH and EC at study termination, measured with the 2:1 dilution method, in sub-irrigated and drip-irrigated tomato. Mean ± standard error of the mean.

| Nutrient Solution EC dS m$^{-1}$ | Substrate pH * | Substrate EC dS m$^{-1}$ |
|---|---|---|
| 1.4 | 6.55 ± 0.08 a | 1.21 ± 0.09 c |
| 2.0 | 6.25 ± 0.04 b | 1.59 ± 0.15 b |
| 2.4 | 6.05 ± 0.03 c | 1.94 ± 0.17 a |
| 2.0 (Drip irrigation) | 6.27 ± 0.03 b | 0.88 ± 0.05 d |

* Different letters indicate significant differences according to Tukey´s multiple mean comparison test ($p < 0.05$).

The EC of the substrate in sub-irrigated plants, measured with the data logger Em50 system ECH$_2$O (Software: ECH$_2$O Utility), increased through the growing season as the irrigation events were applied (Figure 6). Figure 6 shows three EC measurements at each irrigation event; the first one corresponds to the EC reading prior to irrigation, the second one corresponds to the peak EC read when the substrate was saturated with the nutrient solution, and the third EC measurement corresponds to the reading when the substrate was at container capacity after drainage ceased. The substrate EC exhibited a notable increase when the ninth irrigation event was applied to plants treated with solutions of 2.4 dS m$^{-1}$, reaching values up to 1.25–1.50 dS m$^{-1}$; however, after the 31st irrigation event, the substrate EC exhibited a second increase, reaching up to 2.00–2.75 dS m$^{-1}$ (Figure 6). Similar trends were observed when the nutrient solutions had an EC of 2.0 dS m$^{-1}$, however, it was up to 0.60–0.70 dS m$^{-1}$ and to 0.80–1.40 dS m$^{-1}$ after the ninth and 31st irrigation, respectively (Figure 6). In contrast, when plants were sub-irrigated with solutions of 1.4 dS$^{-1}$, the first increase in substrate EC was observed after the 16th irrigation (0.10–0.20 dS m$^{-1}$) while the second increase was observed after the 32nd irrigation (0.25–0.60 dS m$^{-1}$) (Figure 6). The increase in EC has been associated with salt accumulation throughout the growing season due to zero leaching, as reported by several authors [27,30,46]. Similar results were reported in sugar liners as using solutions of higher nutrient concentration resulted in a small reduction in substrate EC, however, after day 21 of treatment, the EC increased [47].

In contrast with the EC, the pH tended to increase in plants sub-irrigated with solutions of lower EC (Table 1). Comparable reports were described for sugar cane (*Saccharum* sp.) liners grown in an ebb and flow sub-irrigation system, as the substrate pH tended to decrease after nine days of treatments [47]. The decrease in substrate pH has been attributed to the displacement of protons by the higher cation concentration of the nutrient solutions [48].

These results indicate that substrate EC increased according to the EC of the nutrient solution used in sub-irrigated plants, however, the substrate EC increased more rapidly when the nutrient solutions were of 2.0 and 2.4 dS m$^{-1}$ (at the ninth and 31st irrigation) than when the solutions were of 1.4 dS m$^{-1}$ (at the 16th and 32nd irrigation). This suggests that a strategy to reduce the impact on substrate EC would focus on reducing the EC of the nutrient solution from 2.0 to 1.4 dS m$^{-1}$ prior to the 31st irrigation.

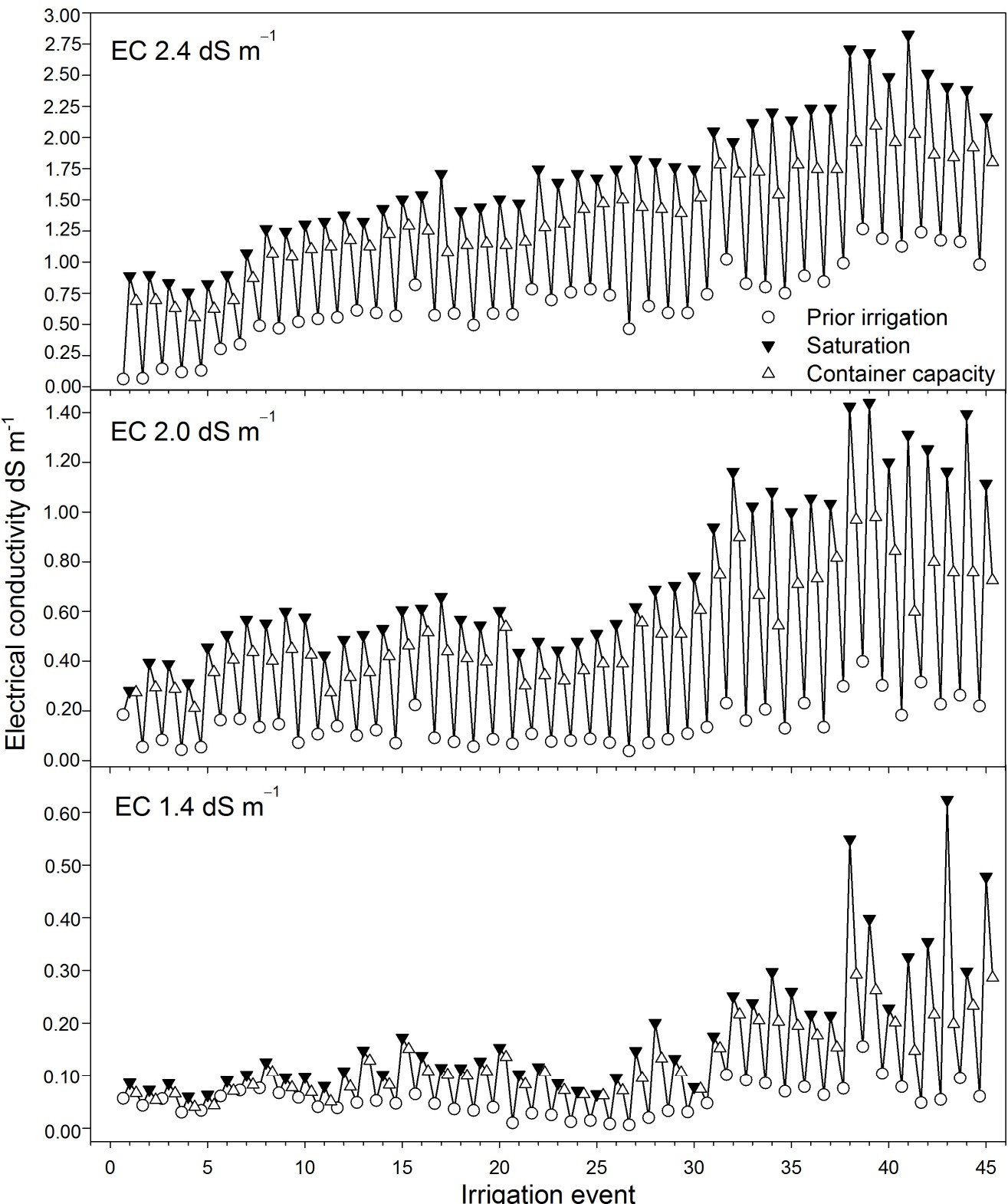

**Figure 6.** Electrical conductivity (EC) on the substrate, recorded with a data logger Em50 system ECH$_2$O (Software: ECH$_2$O Utility), on 45 irrigation events as affected by the EC of the nutrient solutions used for the sub-irrigation of tomato plants. EC was recorded at three moments: prior to each irrigation event, at the time when the substrate was completely saturated with the nutrient solution, and at container capacity once drainage was ceased. Data for drip-irrigated plants were not measured.

The present study also demonstrates that substrate EC, measured at container capacity after sub-irrigation events, showed a relationship with the yield of the respective truss (Figure 7). In general, the highest yield was obtained with the lowest substrate EC for each nutrient solution; however, as salts built up through continuous irrigation there was a reduction in fruit productivity. These results show that for maximum production of sub-irrigated tomato, the substrate EC should be maintained at about 0.12 dS m$^{-1}$ when the nutrient solution has an EC of 1.4 dS m$^{-1}$, whereas it should be at 0.47 and 1.5 dS m$^{-1}$ when the nutrient solution has an EC of 2.0 and 2.4 dS m$^{-1}$, respectively.

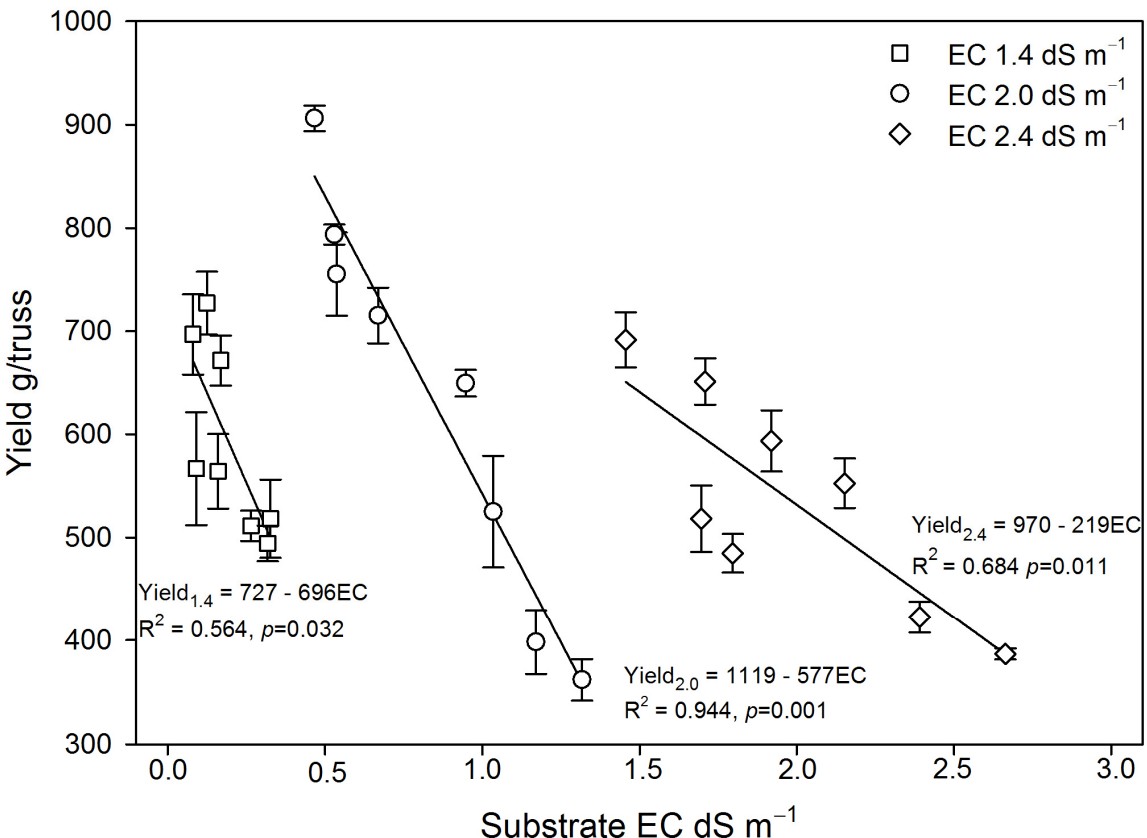

**Figure 7.** Relationship between the substrate electrical conductivity (EC) and yield at harvest time of each of eight trusses in tomato plants sub-irrigated with nutrient solutions of different EC. Substrate EC was the average of the EC measured at container's capacity recorded with a data logger Em50 system ECH$_2$O (Software: ECH$_2$O Utility) when the respective truss was harvested.

## 4. Conclusions

The results of this study allow us to conclude that sub-irrigation can be applied successfully for the production of tomatoes with no deleterious effects on fruit yield as long as the EC of the nutrient solution is carefully managed. A steady increase in substrate EC occurs as the growing season progresses due to salts accumulation, therefore, in order to avoid negative impacts on fruit yield, the EC of the nutrient solution should be decreased to achieve maximum fruit production. The sub-irrigation of tomato with nutrient solutions of 2.0 dS m$^{-1}$ proved to produce the highest yields from the start of the growing season up to the harvest of the fourth truss; however, prior to harvesting the last two trusses, the EC of the nutrient solution should be decreased to 1.4 dS m$^{-1}$ in order to maintain the high yield and fruit size. The substrate EC increased rapidly when the nutrient solutions were of 2.0 and 2.4 dS m$^{-1}$ since bursts in EC were detected at the 9th and 31st irrigation, whereas with solutions at 1.4 dS m$^{-1}$ the burst occurred later, at the 16th and 32nd irrigation, suggesting that a good strategy would reduce the EC of the nutrient solution from 2.0 to 1.4 dS m$^{-1}$

prior to the 31st irrigation. The higher fruit yield of sub-irrigated plants was associated with a decreased dry weight of the vegetative plant parts, suggesting that sub-irrigation is associated with a more favorable partitioning of biomass towards the fruits. The size of the fruits harvested from plants treated with solutions of 2.0 dS m$^{-1}$ was comparable to that obtained by the drip-irrigated plants; however, in order to obtain fruits of better size at the final harvests, decreasing the EC of the nutrient solution to 1.4 dS m$^{-1}$ would also improve the quality of fruits.

**Author Contributions:** Data curation, L.A.V.-A.; Formal analysis, M.C.-Z. and J.A.G.-F.; Funding acquisition, L.A.V.-A.; Investigation, A.M.-C. and L.A.V.-A.; Methodology, A.M.-C. and D.A.-C.; Project administration, L.A.V.-A.; Resources, D.A.-C. and J.A.G.-F.; Supervision, L.A.V.-A. and D.A.-C.; Visualization, D.A.-C.; Writing—original draft, A.M.-C. and L.A.V.-A.; Writing—review and editing, M.C.-Z. and J.A.G.-F. All authors have read and agreed to the published version of the manuscript.

**Funding:** This research received no external funding.

**Data Availability Statement:** Not applicable.

**Acknowledgments:** The authors thank the National Council of Science and Technology of México (CONAHCYT) for supporting Ariel Méndez-Cifuentes' Scholarship and the Universidad Autónoma Agraria Antonio Narro for supporting this research.

**Conflicts of Interest:** The authors declare no conflict of interest.

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
