# Peer review of "Nutrient Solution Electrical Conductivity Affects Yield and Growth of Sub-Irrigated Tomatoes"

_horticulturae, doi:10.3390/horticulturae9070826_

Round 1
Reviewer 1 Report
Review of the article " Electrical Conductivity of the Nutrient Solution Affects Yield 2
and Growth of Sub-Irrigated Tomato in Containers" In the journal "Horticulture”.
In today's environment of progressive pollution environment, the need to save water and electricity production of fresh vitamin ecologically clean vegetable production on multi-storey units is an urgent task.
In the introduction, the authors provide references to 23 literary sources, which is 62% of the literature used in the article.
The aim of the research is formulated.
In the section of materials and methods, the conditions of the experiment of growing tomatoes are described in detail.
The results of the experiment are illustrated with 5 figures.
The authors are recommended to pay serious attention to the following:
1. In the Abstracts (line 28-29) it is
«Dry weight of vegetative plant parts was significantly reduced in sub-irrigation 28
plants compared to drip-irrigated plants, suggesting a shift in dry mass partitioning. Our results 29.»
However, in the description of the results this information is missing. Also this issue is not reflected in the conclusion.
2. Remarks to the section Materials and methods:
- it is necessary to write clearly the period of observation of growth and development of tomato plants and years of research;
- was the room ventilated and at what temperature?
- in point 2.2 to specify how the pH and EC of the substrate were regulated;
- during the growing cycle was the secondary pruning of the stems carried out? If yes, at what time?
- Were measures taken to promote fruit setting (e.g. bumblebees)?
- Were the growth patterns of the plants recorded under experimental conditions - shoot formation, length, number of leaves, formation of lateral shoots and inflorescences, ovary formation? In addition to recording the weight of biomass, it is necessary to give data on the growth of tomato plants;
-Did you take into account fruit size when counting the yield? Data on tomato marketability classes should be given;
- It would be very advisable to estimate at least general indicators of fruit quality, since the absence of the quality characteristics of the obtained fruits significantly reduces the value of the work performed.
3. Remarks to the section Results and discussion:
- it would be more correct to initially describe the character of growth and development of tomato plants, and then talk about the yield;
- how do you explain the effect of experiment variants on yield, if there are no differences in plant biomass changes? This question needs to be carefully worked out. Carefully read the sentences of lines 219 - 228 and eliminate the contradictions. Conduct a thorough analysis of your results.
4. Comments on section 5, you need to eliminate contradictions:
- How can higher yields be related to decreased dry matter of vegetative parts of plants lines 315-316.
The statistical analysis is compelling.
Once the shortcomings are corrected, the article will meet the rating of Horticulture magazine can be recommended for publication.
Author Response
|
REVIEWER´S COMMENTS |
AUTHOR´S COMMENTS |
ABSTRACT |
In the Abstracts (line 28-29) it is «Dry weight of vegetative plant parts was significantly reduced in sub-irrigation plants compared to drip-irrigated plants, suggesting a shift in dry mass partitioning. Our results» However, in the description of the results this information is missing. Also this issue is not reflected in the conclusion |
The following sentence clarifying this statement was included in the 3.3 Biomass production section: “These results indicate that the unaffected fruit production in sub-irrigated plants was associated with a decrease in biomass of the vegetative plants parts, suggesting that under sub-irrigation there is a tendency to divert the biomass from vegetative organs towards fruit formation” In the conclusions, Lines – we wrote: “The higher fruit yield of sub-irrigated plants was associated to a decreased dry weight of the vegetative plant parts, suggesting that sub-irrigation is associated with a more favorable partitioning of biomass towards the fruits.” |
MATERIALS AND METHODS |
2. Remarks to the section Materials and methods: A- it is necessary to write clearly the period of observation of growth and development of tomato plants and years of research;
B- was the room ventilated and at what temperature?
C- in point 2.2 to specify how the pH and EC of the substrate were regulated;
D- during the growing cycle was the secondary pruning of the stems carried out? If yes, at what time?
E- Were measures taken to promote fruit setting (e.g. bumblebees)?
F- Were the growth patterns of the plants recorded under experimental conditions - shoot formation, length, number of leaves, formation of lateral shoots and inflorescences, ovary formation?
G-In addition to recording the weight of biomass, it is necessary to give data on the growth of tomato plants;
H-Did you take into account fruit size when counting the yield? Data on tomato marketability classes should be given;
I- It would be very advisable to estimate at least general indicators of fruit quality, since the absence of the quality characteristics of the obtained fruits significantly reduces the value of the work performed.
|
A. The following sentence was added to the manuscript: “Seedlings of hybrid tomato cv Climstar were transplanted on August 7th, 2018 and the experiment was concluded on January 11th, 2019”
B. The following sentence was added to the manuscript: “The greenhouse had a fan and pad cooling system with a temperature set point of 25.0 ºC and a heater unit set at 10 ºC”
C. pH and EC were not controlled throughout the experiment as one of the objectives of the study was to determine the effect of the EC of the nutrient solutions on the final pH and EC of the substrate
D. The following sentences were added: “Plants were trellised to one stem while the leaves were pruned periodically throughout the study period to maintain 11 to 13 mature leaves; eight trusses were allowed to develop and they were pruned to maintain five flowers each.”
E. We did not use bumblebees nor growth regulators to promote fruit production, pollination of flowers was granted with the air movement when the fan and pad system was working.
F. The plants were pruned to allow the development of only one stem so that side shoots were removed periodically from the main stem. The following sentences was added: “Plants were trellised to one stem by removing the side shoots, while the leaves were pruned periodically throughout the study period to maintain 11 to 13 mature leaves; eight trusses were allowed to develop and they were pruned to maintain five flowers each.”
G. Unfortunately We do not have information about growth rates.
H. As suggested by the Reviewer, a new section showing the effects of the treatments on fruit weight ranked according to US standards
I. The objective of the present study was to determine the effect of sub-irrigation on total fruit yield. We added a new section to the manuscript showing the effects on fruit quality measures as the size of the fruits |
RESULTS AND DISCUSSION |
A- it would be more correct to initially describe the character of growth and development of tomato plants, and then talk about the yield;
B- how do you explain the effect of experiment variants on yield, if there are no differences in plant biomass changes? This question needs to be carefully worked out.
C-Carefully read the sentences of lines 219 - 228 and eliminate the contradictions.
D- Conduct a thorough analysis of your results.
|
A- Suggestion accepted, the change was incorporated
B-In current Figure 1 (Figure 3 in the last version) we show that there was a detrimental effect of subirrigation on plant growth as biomass of leaves and stem was reduced. The effect on yield is explained by the lower substrate EC achieved by plants treated with solutions of 2.0 dS/m; however, this EC was not low enough to impair the yield as it happened when the solution had 1.4 dS/m. This information is in Figure 5. We think that the explanation requested by the Reviewer is on lines 235-246.
C-We regret the confusion. On these lines, we are describing the effects on substrate EC as affected by the EC of the nutrient solutions, and then associating such increases in substrate EC with the decreasing effect on fruit yield.
D- We revised the discussion as suggested by the Reviewer |
CONCLUSIONS |
A- Comments on section 5, you need to eliminate contradictions:
B- How can higher yields be related to decreased dry matter of vegetative parts of plants lines 315-316.
|
A- The Conclusions section was revised and now it reads as follows: “The results of this study allow us to conclude that sub-irrigation can be applied successfully for the production of tomatoes with no deleterious effects on fruit yield as long as the adequate EC of the nutrient solution is used. A steady increase in substrate EC occurs along the growing season due to salts accumulation; therefore, in order to avoid negative impacts on fruit yield, the EC of the nutrient solution should be decreased to achieve maximum fruit production. The sub-irrigation of tomato plants with nutrient solutions at ECs of 2.0 dS m-1 proved to produce the highest yields from the start of the growing season up to the harvest of the fourth truss, however, previous to harvesting the last trusses, the EC of the nutrient solution should be decreased to 1.4 dS m-1 in order to maintain the high yield. The higher fruit yield of sub-irrigated plants was associated with a decreased dry weight of the vegetative plant parts, suggesting that sub-irrigation is associated with a more favorable partitioning of biomass towards the fruits”
B- As previously discussed, this may be due to a shift in the allocation of dry matter from the vegetative to the generative plant parts. In tomato, growers are compelled to avoid excessive growth of the vegetative plant parts as this is reflected in the growth of the fruits as long as there is a proper vegetative/generative balance. The conclusion section contains a line (The higher fruit yield of sub-irrigated plants was associated with a decreased dry weight of the vegetative plant parts, suggesting that sub-irrigation is associated with a more favorable partitioning of biomass towards the fruits) that describes this hypothesis |

Reviewer 2 Report
It is a well prepared manuscript. There are some minor corrections in the text. The conclusion (last sentence) "The fruit yield of sub-irrigated plants.... has to be better explained and cited in the corresponding text.

Author Response
REVIEWER 2
|
REVIEWER´S COMMENTS |
AUTHOR´S COMMENTS |
ABSTRACT |
It is a well prepared manuscript. There are some minor corrections in the text. The conclusion (last sentence) "The fruit yield of sub-irrigated plants.... has to be better explained and cited in the corresponding text.
|
We appreciate the comments by Reviewer. The last sentence was |
L68. |
Is the introduction cited authors in another font style, it is correct? |
Yes, the format fits the journal´s demands as to citing references style |
L245 |
Incorrect “-“ |
The symbol was changed into the superscript format |
L258-261 |
The has to be better explained and cited in the corresponding text |
In L193-198 we wrote: “The fact that plants sub-irrigated with nutrient solutions of 1.4 and 2.0 dS m-1 exhibited a yield comparable to the of the drip-irrigated plants despite they had a lower biomass accumulation suggests that tomato partitioning of reserves were shifted towards fruit production on these plants; similar trends have been also reported by Méndez-Cifuentes et al. [20]. According to Ji et al. [31] dry mass partitioning to tomato fruits may be associated with an increased fruit sink strength due to an enhanced transport and metabolism of sugars, which in turn may be related with an improved nutrient balance”
|

Reviewer 3 Report
1) The title should be changed to THE INFLUENCE OF ELECTRICAL CONDUCTIVITY OF THE NUTRIENT SOLUTION ON YIELD AND GROWTH OF SUB-IRRIGATED TOMATO IN CONTAINERS.
2) The format of Abstract is not according to the MDPI journals.
3) Please, re-write the abstract, some words have repeated frequently and some sentences are not clear, for example however, in life 16 and 22, please try to add other words or re-write the Abstract to be more attractive for readers.
4) The authors have used the word (( efficient )) so much in the manuscript like Abstract and Introduction (line 41), please substitute this word with new words.
5) Line 43, according to (6), write the name of authors and then add the Reference number of re-write it and add (6) in the end of sentence.
6) Line 61, as reported by (4), (13), (14) and (15), please, re-write the sentence and write in the way that you put all References in the end of sentence.
7) The introduction has written very well, but authors have not used Paragraphing in an appropriate way, each paragraph should start with new concepts, not just make a new paragraph regularly.
8) Line 171, Cucurbita pepo L., it seems the format of this word is different from other sentences, please, check it out.
9) In Figure 1, authors have used scientific name of tomato in the sentence (Solanum lycopersicum L.), just one time using scientific words for plants is enough, when and if the authors use the Scientific name, they do not need to repeat it again.
10) This is also true for line 229, pumpkin (Curcurbita pepo L.). authors just need to use the scientific name of plants and crops once in the article.
11) Again, in Results and Discussion, paragraphing is not correct.
12) Please check the format of Table, the titles should BOLD (Nutrient solution EC; Substrate pH; Substrate EC).
13) Please, re-design and re-write the Conclusions part, it is too short, please, clearly explain the results and give your suggestions and recommendations for future researches.
14) Open one part for Abbreviations after Conclusions before References.
15) Please, check the format of all references, for example in Reference 1, it should Mojid, M.A.; NOT Mojid M. A.; (no space between M and A).
16) Increase Discussion part by adding more and new references, it seems 37 References for this topic is not enough, try to use more updated and new references in Discussion part.
The manuscript just needs MINOR English revision.
Author Response
REVIEWER´S COMMENTS |
AUTHOR´S COMMENTS |
1) The title should be changed to THE INFLUENCE OF ELECTRICAL CONDUCTIVITY OF THE NUTRIENT SOLUTION ON YIELD AND GROWTH OF SUB-IRRIGATED TOMATO IN CONTAINERS. |
The suggestion was accepted |
2) The format of Abstract is not according to the MDPI journals |
The abstract was reduced to 200 words |
3) Please, re-write the abstract, some words have repeated frequently and some sentences are not clear, for example however, in life 16 and 22, please try to add other words or re-write the Abstract to be more attractive for readers. |
The abstract was reviewed and repeated words was reduced |
4) The authors have used the word (( efficient )) so much in the manuscript like Abstract and Introduction (line 41), please substitute this word with new words. |
The word “efficient” appears only two times in the manuscript, in Lines 16 and 39. It was changed to “effective” on Line 39. The word “efficiency” now appear only 3 times in the manuscript. |
5) Line 43, according to (6), write the name of authors and then add the Reference number of re-write it and add (6) in the end of sentence. |
Sorry for the mistake, it was corrected |
6) Line 61, as reported by (4), (13), (14) and (15), please, re-write the sentence and write in the way that you put all References in the end of sentence. |
The references were changed to [4, 13–15] according to journal´s standards |
7) The introduction has written very well, but authors have not used Paragraphing in an appropriate way, each paragraph should start with new concepts, not just make a new paragraph regularly. |
Paragraphing in the Introduction was revised as suggested by the Reviewer |
8) Line 171, Cucurbita pepo L., it seems the format of this word is different from other sentences, please, check it out. |
The font style was changed |
9) In Figure 1, authors have used scientific name of tomato in the sentence (Solanum lycopersicum L.), just one time using scientific words for plants is enough, when and if the authors use the Scientific name, they do not need to repeat it again. |
Solanum lycopersicum was eliminated from figures and tables and now it appears only once in the materials and methods section |
10) This is also true for line 229, pumpkin (Curcurbita pepo L.). authors just need to use the scientific name of plants and crops once in the article. |
It was eliminated in the second time this scientific name was used |
11) Again, in Results and Discussion, paragraphing is not correct. |
Paragraphing in the Introduction was revised as suggested by the Reviewer |
12) Please check the format of Table, the titles should BOLD (Nutrient solution EC; Substrate pH; Substrate EC). |
The font on the titles was changed to bold |
13) Please, re-design and re-write the Conclusions part, it is too short, please, clearly explain the results and give your suggestions and recommendations for future researches. |
The conclusions section was revised as suggested by the Reviewer |
14) Open one part for Abbreviations after Conclusions before References. |
According to Journal´s style adding a section to abbreviations is not necessary: Acronyms/Abbreviations/Initialisms should be defined the first time they appear in each of three sections: the abstract; the main text; the first figure or table. When defined for the first time, the acronym/abbreviation/initialism should be added in parentheses after the written-out form. |
15) Please, check the format of all references, for example in Reference 1, it should Mojid, M.A.; NOT Mojid M. A.; (no space between M and A). |
The mistake was corrected in all the references |
16) Increase Discussion part by adding more and new references, it seems 37 References for this topic is not enough, try to use more updated and new references in Discussion part. |
The discussion section was increased adding 10 more references related to the topics of the study |
Comments on the Quality of English Language. The manuscript just needs MINOR English revision. |
The English language was reviewed |
Reviewer 4 Report
This manuscript entitled “Electrical Conductivity of the Nutrient Solution Affects Yield and Growth of Sub-Irrigated Tomato in Containers” used sub-irrigation to cultivate tomatoes. Different ECs (1.4, 2.0, and 2.4 dS m-1) of the nutrient solution were applied. The manuscript concluded that sub-irrigation was successfully applied to tomato cultivation and the EC of 2.0 dS m‑1 was the best. This manuscript fits the scope of the journal and sheds some light on tomato production. But some obvious shortages need to be addressed before acceptance.
Major concerns:
--The language needs polishment and can be more concise. Some sentences are hard to read, such as Lines 169-174.
--Introduction: The introduction section is a bit confusing. The introduction is the background of the study. The authors should tell the readers what is sub-irrigation, why we need to care about sub-irrigation, and why the EC value of the nutrient solution is important in sub-irrigation. In this way, the readers can understand the aim of the study. Therefore, the introduction section needs substantial revision.
--2.4 Electrical conductivity treatments: why used 1.4, 2.0, and 2.4 dS m-1? Is it from the previous study or the preliminary experiment? The relevant information should be added here.
--Why the authors didn’t determine the EC values or the osmotic regulation substances in the tomato plants, especially the roots? These indexes are important for understanding this article.
Minor:
Line 140 ºC--°C;
Line 157 p should be in italics;
Line 269 1.4 dS m-1;
The language needs polishment and can be more concise. Some sentences are hard to read, such as Lines 169-174.
Author Response
REVIEWER´S COMMENTS |
AUTHOR´S COMMENTS |
--Introduction: The introduction section is a bit confusing. The introduction is the background of the study. The authors should tell the readers what is sub-irrigation, why we need to care about sub-irrigation, and why the EC value of the nutrient solution is important in sub-irrigation. In this way, the readers can understand the aim of the study. Therefore, the introduction section needs substantial revision. |
A description of the sub-irrigation system was included in the introduction. The paragraph added reads as follows: “According to Ferrarezi et al. [2015], sub-irrigation is a technique that provides the fertilizer solution to the bottom of containers and by capillary action water and nutrients are provided to the roots through holes located in the container; in this system, container-grown plants are periodically flooded with the nutrient solution using a closed loop system and that, in case that there is an excess on the nutrient solution, it is collected and reused for subsequent irrigation events.” We think that the issues about the importance of EC and the sub-irrigation system are already addressed in the introduction |
--2.4 Electrical conductivity treatments: why used 1.4, 2.0, and 2.4 dS m-1? Is it from the previous study or the preliminary experiment? The relevant information should be added here. |
The ECs included in the treatments are based on standard values used in the industry, tomato is reported to grow in a rank from 2.0 to 2.5 dS/m. In the discussion we included a new reference indicating that ECs higher than 3.2 dS/m are detrimental for tomato. |
--Why the authors didn’t determine the EC values or the osmotic regulation substances in the tomato plants, especially the roots? These indexes are important for understanding this article. |
We regret we did not include in our study the determination of osmolytes in tomato roots, but definitely they are an important parameter to be determined in future experiments |
Line 140 ºC--°C |
The manuscript was revised to change the symbol for centigrades |
Line 157 p should be in italics; |
“p” was converted to italics |
Line 269 1.4 dS m-1; |
The manuscript was revised to use the superscript |
The language needs polishment and can be more concise. Some sentences are hard to read, such as Lines 169-174. |
The manuscript was revised for language mistakes and readability. The sentence indicated by the Reviewer now reads: “These results indicate that in order to avoid yield reduction over time in sub-irrigated tomato, the concentration of the nutrient solution must be adjusted as the growing season progresses. García-Santiago et al. [16] reported similar tendencies as during the first month of harvest, sub-irrigated tomatoes rendered the highest yields when the nutrient solution had en EC of 1.6 and 2.4 dS m-1, while from the fourth to the sixth month, the higher yields were from plants treated with nutrient solution at 1.2 dS m-1. On the first, third, fifth, sixth and seventh truss there were no effects on yield among ECs (Figure 3). Fayezizadeh et al. [2021] reported similar results as fruits from the first to the third cluster in tomato cv. V4-22 produced yields from 548 to 634 g in a close-loop irrigation system, however, the yield decreased gradually up to 394 g by the seventh cluster. “
|
Round 2
Reviewer 4 Report
The authors have addressed all my concerns. The manuscript has been improved a lot. I recommend accepting this article in the current version.